# The Effects of Gluconacin on Bacterial Tomato Pathogens and Protection against *Xanthomonas perforans*, the Causal Agent of Bacterial Spot Disease

**DOI:** 10.3390/plants12183208

**Published:** 2023-09-08

**Authors:** Elizabeth Teixeira de Almeida Ramos, Fábio Lopes Olivares, Letícia Oliveira da Rocha, Rogério Freire da Silva, Margarida Goréte Ferreira do Carmo, Maria Teresa Gomes Lopes, Carlos Henrique Salvino Gadelha Meneses, Marcia Soares Vidal, José Ivo Baldani

**Affiliations:** 1Programa de Pós-Graduação em Fitotecnia (PPGF), Departamento de Fitotecnia, Instituto de Agronomia, Universidade Federal Rural do Rio de Janeiro, Rodovia BR 465, km 07, Seropédica 23890-000, RJ, Brazil; elizabethramos_vr@hotmail.com (E.T.d.A.R.); gorete.carmo1@gmail.com (M.G.F.d.C.); 2Núcleo de Desenvolvimento de Insumos Biológicos para a Agricultura, Laboratório de Biologia Celular e Tecidual, Centro de Biociências e Biotecnologia, Universidade Estadual do Norte Fluminense Darcy Ribeiro, Campos dos Goytacazes 28013-602, RJ, Brazil; fabioliv@uenf.br (F.L.O.); leticiarocha2004@gmail.com (L.O.d.R.); 3Programa de Pós-Graduação em Ciências Agrárias, Centro de Ciências Biológicas e da Saúde, Departamento de Biologia, Universidade Estadual da Paraíba, Universitário, Campina Grande 58429-500, PB, Brazil; rogerio.freire.silva@aluno.uepb.edu.br (R.F.d.S.); carlos.meneses@gsuite.uepb.edu.br (C.H.S.G.M.); 4Faculdade de Ciências Agrárias, Universidade Federal do Amazonas, Avenida Rodrigo Otávio Ramos, 3.000, Bairro Coroado, Manaus 69077-000, AM, Brazil; mtglopes@ufam.edu.br; 5Embrapa Agrobiologia, Rodovia BR 465, km 07, Seropédica 23891-000, RJ, Brazil; marcia.vidal@embrapa.br

**Keywords:** biological control, bacteriocin, *G. diazotrophicus*, bacterial spot, *Solanum lycopersicum* L.

## Abstract

As agricultural practices become more sustainable, adopting more sustainable practices will become even more relevant. Searching for alternatives to chemical compounds has been the focus of numerous studies, and bacteriocins are tools with intrinsic biotechnological potential for controlling plant diseases. We continued to explore the biotechnological activity of the bacteriocin Gluconacin from *Gluconacetobacter diazotrophicus*, PAL5 strain, by investigating this protein’s antagonism against important tomato phytopathogens and demonstrating its effectiveness in reducing bacterial spots caused by *Xanthomonas perforans*. In addition to this pathogen, the bacteriocin Gluconacin demonstrated bactericidal activity in vitro against *Ralstonia solanacearum* and *Pseudomonas syringae* pv. *tomato*, agents that cause bacterial wilt and bacterial spots, respectively. Bacterial spot control tests showed that Gluconacin reduced disease severity by more than 66%, highlighting the biotechnological value of this peptide in ecologically correct formulations.

## 1. Introduction

Protecting crops from plant diseases is essential to meeting the current demand for food quantity and quality required globally. Crop pests and pathogens are widely recognized as significant obstacles to the balance of agricultural production systems, generating productivity losses of 40% [1].

For example, the tomato (*Solanum lycopersicum* L.—Solanaceae family) is one of the most affected crops due to its susceptibility to pests and diseases [2]. According to the FAO in 2022, the world production of tomatoes was superior to 187 million tons, with more than 5.5 million ha of harvested area [3]. Although yield is expected to increase, the lack of cultivars and products resistant to pests and pathogens promote significant losses in the tomato market [4,5].

Several pests and diseases cause tomato crop and fruit losses. However, bacterial spots have been highlighted as one of the most destructive diseases, causing productivity losses > 50% when tomato plants are grown under the conditions most likely to develop disease [6,7]. This disease has a generalized worldwide distribution and is caused by four *Xanthomonas* species: *X. euvesicatoria* and *X. vesicatoria* (distributed worldwide), *X. gardneri* (originally from Costa Rica and Yugoslavia but has been found in Brazil and USA), and *X. perforans* (USA, Mexico, Thailand, and Brazil) [8]. Copper-based antimicrobial compounds have been used to manage this disease [9]. However, in recent decades, these products have raised concerns over their sustainability and adverse environmental effects [10], particularly variability in protection efficiency [11], dissemination of resistance genes [9,12], and harmful effects on human and animal health [13].

Applying sustainable alternatives for disease control have been increasing exponentially, and its insertion into Integrated Pest Management (IPM) is now frequent. Plant growth-promoting bacteria (PGPB) and biocontrol agents are the focus of such research due to their potential for antibiosis against various phytopathogens and functional characteristics promoting plant growth [14,15].

These beneficial microorganisms may promote biocontrol indirectly by inducing systemic resistance in plants [16] and directly by synthesizing inhibitory allelochemicals such as antibiotics, antifungal metabolites, and bacteriocins [17]. Bacteriocins are bioactive peptides with inhibitory activity against various micro-organisms [18]. Our group has been studying Gluconacin, a bacteriocin produced by *Gluconacetobacter diazotrophicus* strain PAL5. This antimicrobial peptide has demonstrated inhibitory activity against various sugarcane pathogens [19] and stability at high temperatures and acidic pH [20]. Therefore, it shows promising characteristics for ecological formulations and new sustainable agricultural practices. We evaluated Gluconacin’s inhibitory effects against tomato phytopathogens in vitro and validated its biotechnological potential in protecting tomato plants inoculated with *Xanthomonas perforans*, the causal agent of bacterial spot disease, under greenhouse conditions.

## 2. Results

### 2.1. In Vitro Assay Demonstrates the Anti-Bacterial Effects of Gluconacin against Phytopathogenic Strains

The bacterial counting assay revealed that the purified bacteriocin Gluconacin exhibited high antibacterial activity against the phytopathogenic bacterial strains tested (Figure 1). *R. solanacearum* demonstrated greater sensitivity to the antimicrobial peptide, with growth totally inhibited when incubated with 0.5 µg·µL^−1^ of Gluconacin. For the *X. perforans* and *P. syringae* pv. *tomato* strains, the MIC of Gluconacin required was 1.0 µg·µL^−1^. *X. vasicola* pv. *vasculorum* and *X. albilineans* strains required a lower Gluconacin MIC (0.25 µg·µL^−1^) (Appendix A).

### 2.2. Gluconacin Successfully Controls the Infection of Tomato Plants Inoculated with Xanthomonas perforans

The characteristic symptoms of bacterial spot disease were observed seven days after inoculation (DAI). They first appeared on the older leaves in necrotic brown spots with yellowish halos (Figure 2A). Ten days after inoculation, some plants showed detachment from the necrotic area, resulting in perforations (Figure 2B) and yellowing of the stem. Sixteen days after infection, intensification of chlorosis, coalescence of perforations, and the falling of leaves were observed (Figure 2C). These symptoms were much more evident in inoculated plants not treated with Gluconacin. Plants treated with Gluconacin showed delayed symptoms, which were milder than untreated plants (Figure 3 and Figure 4).

The disease severity in plants treated with Gluconacin reduced by 66.4%, as shown in the area below the disease progressive curve (Figure 4). As expected, treating plants with elution buffer (Control I) and saline solution (Control II) did not reduce bacterial spot progress, both treatments were statistically similar (Table 1). Treatment with Gluconacin was statistically different from controls, proving to be effective in reducing bacterial spot progress.

### 2.3. Assays by Scanning Electron Microscopy (SEM)

Scanning electron micrographs of tomato leaf samples (24 h after spray) showed that *X. perforans* cells adhered to the periclinal cell wall surface (Figure 5A). After 72 h, we could observe bacteria cells attached to the epidermal cell wall near the stomata complex (Figure 5C). Spraying plants with Gluconacin significantly reduced the pathogen’s colonization of leaf surfaces 24 and 72 h after inoculation. The bacterial infection population reduction was notable just 24 h after inoculation (Figure 5B) and became even more evident after 72 h (Figure 5D).

### 2.4. Bacterial Spot of Tomato Fruits Caused by Xanthomonas perforans Is Controlled by Gluconacin

*X. perforans* induce bacterial spots in tomato fruits, impairing fruit growth and leading to subsequent rot (Figure 6C,F). The effect of Gluconacin treatment associated with *X. perforans* inoculation was significant (Figure 6B,E). It nullified the phytopathogenic action of *X. perforans*, compared to negative controls (Figure 6A,D). As shown in Figure 6, Gluconacin treatment suppresses necroses in the inoculated areas, thus allowing the normal development of fruits and preventing rotting.

## 3. Discussion

The increasing population and global climate change exert considerable pressure on natural resources for food production. Plant pests and diseases harm agricultural production and reduce food production for the human population [21]. In a world scenario where 14% of the food produced is lost in pre-commercialization stages [22], searching for more sustainable agricultural practices is crucial [23]. More than 900 synthetic and natural antimicrobial peptides have been characterized and reported in the literature as potential strategies for agricultural use [24]. Bacteriocins have been highlighted for their versatile application in agriculture, performance against several phytopathogens [25], biostimulation of plant growth [26], and protection against abiotic stresses [27]. Agrochemical compounds’ conventional use in disease and pest control has increased considerably, but consumer acceptance and desire for foods free from pesticide residues continues to grow [28].

Plant growth-promoting bacteria (PGPB) are critical to crop production because they provide the host plant with numerous benefits such as growth stimulation, biofertilization, and protection against pathogenic organisms [29]. Although PGPB’s direct effects have been well-studied, there are still many gaps in knowledge about its indirect benefits, such as disease suppression. The gaps are mainly related to bacteriocins’ protective effect on plants [30]. These biomolecules have an intrinsic agricultural and biotechnological potential, but they have not yet been studied extensively. Therefore, their use has been underestimated [25,31]. According to Grinter et al. [25], Gram-negative bacterial plant pathogens are responsible for various plant diseases, including leaf spots and bacterial wilt. For example, *X. oryzae* pv. *oryzae* is the causal agent of bacterial blight in rice, while other Xanthomonas species can cause diseases with analogous symptoms in maize, sugarcane, and plantain. In addition, other gram-negative bacterial plant pathogens such as *Pseudomonas syringae* have a broad host range and can cause various plant symptoms, including leaf spots, cankers, and wilt. However, the direct application of antibacterial compounds as biological agents in plants has been poorly explored [32].

Despite the lack of research on in vivo bacteriocin application, some results have demonstrated their potential as sustainable tools in plant protection. Various bacteria are known to produce bacteriocins. *Bacillus clausii* GM17 and BacGM17: *B. clausii* GM17 produces a bacteriocin called BacGM17, which inhibits the growth of *Agrobacterium tumefaciens*, a bacterium known for causing crown gall disease in plants [33]. *Bacillus thuringiensis* subsp. *kurstaki* Bn1 and Thuricin Bn1: *B. thuringiensis* subsp. *kurstaki* Bn1 produces thuricin Bn1, which exhibits inhibitory activity against *Pseudomonas savastanoi* and *P. syringae*. These two bacteria are associated with plant diseases [34]. *Bacillus amyloliquefaciens* subsp. *plantarum* FZB42 and Amylocyclicin: *B. amyloliquefaciens* subsp. *plantarum* FZB42 produce a cyclic bacteriocin called amylocyclicin. This bacteriocin has inhibitory activity against certain Gram-positive bacteria, specifically various subspecies of *Clavibacter michiganensis*, which are known to infect the xylem vessels of economically essential host plants [35]. Lactic Acid Bacteria and Nisin: Lactic acid bacteria are another group of bacteria that produce bacteriocins. Nisin, a well-known bacteriocin, is produced by lactic acid bacteria. Nisin has broad-spectrum antimicrobial activity against Gram-positive bacteria, including *Clostridium botulinum*, *Bacillus cereus*, *Listeria monocytogenes*, and *Staphylococcus aureus*. Nisin is often used as a natural food preservative [36]. Our results showed that a consortium of three endophytic lactic acid bacteria, *Lactobacillus plantarum*, *Pediococcus acidilactici*, and *Enterococcus faecium*, exhibited higher antibacterial activity against the pathogen than individual isolates. The endophytic lactic acid bacteria consortium also significantly reduced the incidence of papaya dieback disease in papaya plants under greenhouse conditions. According to Haggag [37], applying bacteriocins (Gramicidin S and polymyxin B bacteriocins extracted from culture filtrate of *Brevibacillus brevis* and *Paenibacillus polymyxa*, respectively) preparation significantly reduced grey mould formation in strawberry plants inoculated with *Botrytis cinerea*. Furthermore, the data variance analysis showed that the concentration of 15 μmol·L^−1^ (Gramicidin S) and 25 μmol·L^−1^ (Polymyxin B) effectively inhibited the multiplication of the pathogen. The experiments also showed that bacteriocins preparation affects the epiphytic survival of the pathogen on treated strawberry plant leaves.

Our team demonstrated the antimicrobial versatility of Gluconacin against various sugarcane phytopathogens. It has high biological stable activity at different temperatures and pH conditions and high bactericidal power after the first seconds of contact [19]. In this study, we elucidated the spectrum of bacteriocin action and verified its potential use against some pathogenic bacteria that attack tomato plants and other relevant crops such as peppers, potatoes, and some species of fruit trees [38,39]. We also evaluated Gluconacin’s effectiveness in controlling bacterial spot disease in tomato plants. We observed that symptoms caused by the pathogen *X. perforans* were alleviated by bacteriocin spraying, reflecting a significant reduction (66%) in disease severity. The scanning electron microscopy (SEM) analysis corroborated these results with a significant reduction in bacterial pathogen colonization. Interestingly, very few bacteria cells were observed attached to the epidermal cell wall surface. Gluconacin treatment likely killed the cells and reduced their adhesion strength, removing moribund cells and debris after SEM sample preparation. Principe et al. [40] also observed a 44% reduction in disease severity of *X. vesicatoria*, another species that causes bacterial spots in tomatoes, when the bacteriocin Tailocin from *P. fluorescens* strain SF4c was used.

The genus *Xanthomonas* has great agronomic relevance and comprises 27 species that cause severe diseases in almost 400 plants (124 monocotyledons and 268 dicotyledons), including a wide variety of important crops such as rice, citrus fruits, tomatoes, and pepper [41,42]. Besides *X. perforans*, we studied the antagonism of Gluconacin against other species of this genus: *X. albilineans*, *X. axonopodis* pv. *Vasculorum,* and *X. vasicola* pv. *vasculorum* [19], demonstrating the broad spectrum of this bacteriocin.

Bacterial spot disease and other diseases caused by *Xanthomonas* sp. are challenging to treat, and copper-based compounds are still the most common form of control [43]. However, several negative points have emerged regarding these chemicals, including environmental impact, the emergence of resistant bacteria, and harmful effects on human and animal health. Consequently, numerous studies have searched for more efficient and sustainable alternatives to copper, and some products have been cited, such as Acibenzolar-S-methyl (ASM) [44], bacteriophages [45], antibiotic streptomycin produced by *Bacillus amyloliquefaciens* strain FZB24 [42], and inducers of systemic acquired resistance (SAR) [46].

The interplay between bacteriocins and microbiomes has proven particularly intriguing, with studies delving into the intricate web of microbial communication. As we unveil how bacteriocins shape the dynamics of these microbial communities, opportunities arise to manipulate ecosystems to improve human health, agricultural production, and innovative ecological approaches. Applying bacteriocins to targeted therapies, specifically aimed at harmful pathogens without impacting beneficial microbiota, offers a promising path to combating infections and diseases. Furthermore, exploring new ecological niches has unveiled a wealth of underexplored bacteriocins with potential applications yet to be fully understood. Diverse ecosystems such as deep-sea marine environments, various soils, and even the human microbiome are being investigated to identify novel sources of bacteriocins with unique characteristics and advantageous properties. As the study of bacteriocins advances, intriguing challenges persist, such as gaining deeper insights into the underlying mechanisms of bacterial resistance to bacteriocin actions. Overcoming these challenges is pivotal to fully harnessing these incredibly versatile molecules’ therapeutic and applicative potential. In an ever-evolving world, the field of bacteriocins continues to inspire the scientific community, offering a captivating glimpse into complex microbial interactions and their promising opportunities.

Gluconacin has great biotechnological potential against tomato bacteriosis, especially in protecting this culture from bacterial spot disease. This study opens new perspectives concerning its application in agriculture. Furthermore, different methods of using bacteriocins, such as encapsulation in nanoparticles [47], sequence engineering for prolongation of shelf antimicrobial activity [48], and transgenic expression in plants [24], have appeared in recent years.

## 4. Materials and Methods

### 4.1. Micro-Organisms and Growth Conditions

The bacterial strains used in the antagonism bioassays, *Ralstonia solanacearum*, *Xanthomonas perforans,* and *Pseudomonas syringae* pv. *tomato* were obtained from the Phytopathogenic Bacteria Collection of the Instituto Biológico de São Paulo and grown in nutrient broth medium (NB—5.0 g·L^−1^ beef peptone; 3.0 g·L^−1^ beef extract; 5.0 g·L^−1^ NaCl, pH 7.0) at 30 °C, 180 rpm for 24 h.

### 4.2. Heterologous Expression and Purification of Gluconacin

We conducted the production and purification of the bacteriocin Gluconacin according to Oliveira et al. [19] with modifications. *Escherichia coli* strain BL21-AI™ was used for heterologous expression, carrying the plasmid p17GDI0415N. Cultivation was performed in lysogeny broth medium (LB—5.0 g·L^−1^ yeast extract, 10 g·L^−1^ NaCl; 10 g·L^−1^ tryptone, pH 7.0) [49] supplemented with 100 μg·mL^−1^ ampicillin at 25°C for 180 rpm. Upon reaching an O.D._600nm_ of 0.4, the culture was added with the expression inducer L-arabinose (0.2%) and incubated at the same temperature for another four hours. The pellet obtained after centrifugation (10,000× *g* rpm, 15 min) was then resuspended in 4 mL of binding buffer (20 mmol·L^−1^ Na_2_HPO_4_ and NH_2_PO_4_, 500 mmol·L^−1^ NaCl, 5 mmol·L^−1^ imidazole, pH 7.4) supplemented with 4 mL of sarcosyl (final concentration 10%).

We purified recombinant Gluconacin using nickel affinity chromatography model His GraviTrap (GE Healthcare, Chicago, IL, USA). We removed proteins loosely bound to the chromatographic matrix through washes with a binding buffer containing 10 and 30 mmol·L^−1^ of imidazole. Elution of the recombinant bacteriocin was made with 4 mL of elution buffer (20 mmol·L^−1^ Na_2_HPO_4_ and NaH_2_PO_4_, 50 mmol·L^−1^ NaCl, 500 mmol·L^−1^ imidazole, pH 7.4). SDS-PAGE monitored the purity of the eluates, and the absorbance determined the quantification at 280 nm through the Beer–Lambert Law.

### 4.3. Antagonistic Bioassays

We evaluated the antimicrobial Gluconacin activity against the phytopathogenic bacteria *X. perforans*, *P. syringae* pv. *tomato*, and *R. solanacearum*. The methodology described by Principe et al. [40] was used for the bacterial strain bioassay.

Three hundred microliters of bacterial culture were transferred to tubes at concentrations ranging from 10^5^ to 10^11^ (colony-forming unit) CFU·mL^−1^ followed by the addition of the same volume of purified Gluconacin (final concentration 0.5 µg·µL^−1^) or elution buffer (negative control). The tubes were incubated at 30°C with shaking at 180 rpm for 24 h. Growth inhibition was analysed by counting viable cells (CFU·mL^−1^) at the bioassay end.

### 4.4. Determination of Minimum Inhibitory Concentration (MIC)

The bacteriocin inhibited bacterial growth and was submitted to a new assay evaluation to determine the MIC. The tests were performed according to De Oliveira et al. [50], with some modifications. The strains were grown to an optical density (OD_600nm_) of 0.5–0.8. Then, the cells were collected by centrifugation (4000× *g* rpm, 15 min) and resuspended in 100 µL of fresh LB medium (~10^8^ CFU·mL^−1^) in a final concentration of (~10^4^ CFU·mL^−1^). Subsequently, the purified Gluconacin was diluted in an elution buffer, and 100 µL of each dilution was added to the bacterial suspension, obtaining the following final concentrations of Gluconacin: 1.0, 0.5, 0.25, 0.125, and 0.06 µg·µL^−1^. The elution buffer was used as a control. The culture was maintained at 30 °C with 180 rpm for 24 h. Bacterial growth was monitored by counting the number of viable cells (CFU·mL^−1^).

### 4.5. Bacterial Spot Disease Control Assay under Greenhouse Conditions

We validated Gluconacin’s antibacterial activity in vivo on tomato plants, specifically the Gaucho cultivar. Initially, we surface-sterilized the tomato seeds with the following solutions: 50% ethanol for 30 s, 0.5% sodium hypochlorite for three minutes, and four washes with sterile distilled water [51]. We placed the seeds in germination trays containing washed sand, Maxplant^®^ commercial substrate and soil (1:1:1) and kept them under greenhouse conditions (28–32 °C) for 15 days.

We sprayed the tomato seedlings (DAG) with Gluconacin solution (1.0 µg·µL^−1^) while applying an elution buffer (Control I) and a saline solution (0.85% NaCl) (Control II) to the controls. One hour later, the seedlings were sprayed with a bacterial suspension containing 10^8^ CFU·mL^−1^ of *X. perforans* strain 2370 in a saline solution. Twenty-four hours before and after inoculation, the plants were kept in a humidification chamber with 70% humidity. Uninoculated plants were used as a negative control. We conducted evaluations 7, 10, 13, and 16 days after infection (DAI) by monitoring the number of lesions in the second pair of leaves (NL). After the end of the assay, the data were integrated over time, and the severity of the bacterial spots was determined from the area under the disease progress curve (*AUDPC*) using the formula [52]:$$AUDPC=\sum_{I=1}^{n-1}[(x_{i}+x_{i+1})/2](t_{i+1}-t_{i})$$
where NL expresses *x* according to the assessment forms described above; *t* signifies time; and *n* signifies the number of assessments over time.

### 4.6. Scanning Electron Microscopy (SEM) Analysis

We collected leaf samples from bacterial spot protection tests for scanning electron microscopy (SEM) analysis. The samples were fixed in 2.5% glutaraldehyde, 4% paraformaldehyde, and 50 mmol·L^−1^ phosphate buffer for 24 h. Subsequently, successive washes were performed for 20 min in the same buffer, followed by dehydration in increasing series of ethanol 30, 50, 70, 90, and 100% for 30 min each. Then, the material was brought to a critical point in CO_2_ using the Critical Point Dryer model CPD030 BAL-TEC (Leica Biosystems, Wetzlar, Germany). An aluminum support was used for assembly, and gold was metallized using a Sputter Coater model SCD050 BAL-TEC (Carl Zeiss AG, Wetzlar, Germany). The analyses were performed using a scanning electron microscope model EVO^®^40 (Jena, Germany) and the second largest city in at 15 Kv.

### 4.7. Effect of Gluconacin on Preventing Bacterial Spot Disease Symptoms in Tomato Fruits

The pellet corresponding to 1.0 mL of *X. perforans* culture, obtained after centrifugation (8000× *g* rpm, 5 min), was resuspended in the same volume of saline solution (0.85% NaCl). We injected 30 µL aliquots of this bacterial suspension (~10^6^ CFU·mL^−1^) into six points of tomato fruit previously disinfected with 70% alcohol. One hour after infection, we injected 30 µL of Gluconacin (1.0 µg·µL^−1^) into the same site. Uninoculated tomatoes (negative control) and tomatoes inoculated with the pathogen and treated with elution buffer (positive control) were used as controls. The systems were maintained at room temperature and observed for 15 days. For each treatment, three tomatoes were used. 

### 4.8. Statistical Analysis

Data were expressed as mean ± standard deviation (SD) and analysed by one-way statistical analysis of variance (ANOVA) followed by Fisher’s Least Significant Difference (LSD) test. All treatments were arranged in a completely randomized design with ten plants for each replication (3n). Data analyses were conducted using SigmaPlot 11.0 software (Systat Software Inc., Chicago, IL, USA). In all cases, differences were considered significant at *p* < 0.05.

## 5. Conclusions

In this manuscript, we demonstrated a promising biotechnological approach for controlling plant diseases, especially tomato pathogens, using the bacteriocin Gluconacin, derived from the PAL5 strain of *G. diazotrophicus*. The protein effectively reduced bacterial spots caused by *X. perforans* and had in vitro bactericidal activity against *R. solanacearum* and *P. syringae* pv. *tomato*. These results suggest that Gluconacin may be a sustainable alternative to chemical compounds for controlling plant diseases. To our knowledge, this is the first report on applying the bacteriocin Gluconacin to tomato plants. More research is needed to evaluate its effectiveness under different conditions and against other pathogens.

## Figures and Tables

**Figure 1 plants-12-03208-f001:**
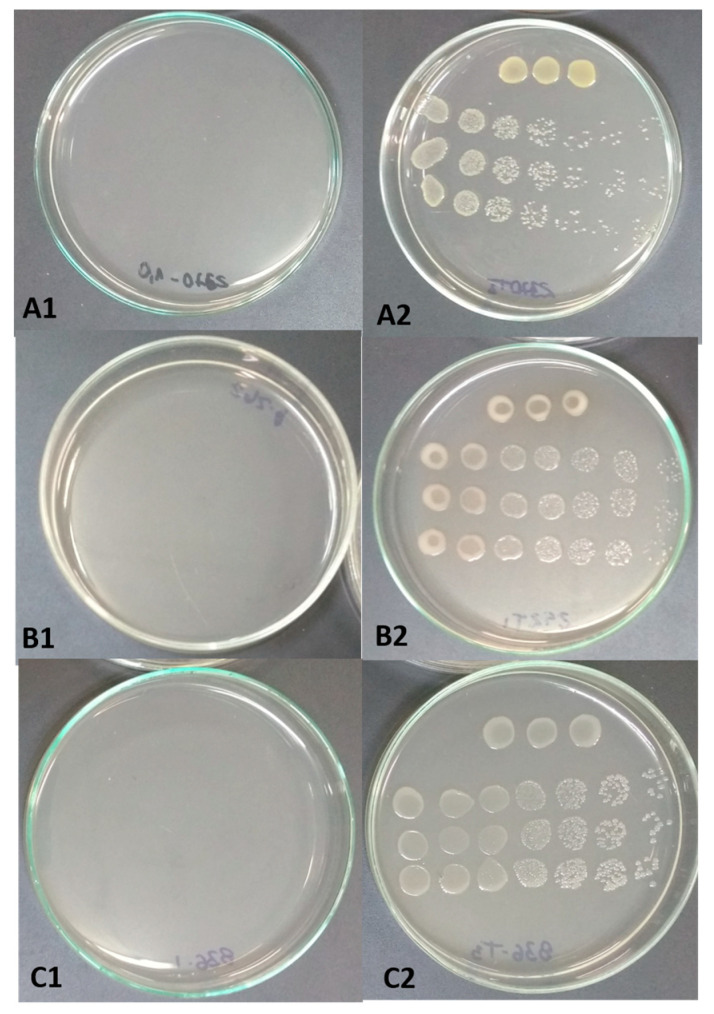
Effects of Gluconacin on tomato phytobacteria. The colony-forming unit number was counted 24 h after incubation with Gluconacin (MIC) (**A1**–**C1**) or Elution buffer (Control) (**A2**–**C2**). (**A1**,**A2**) *X. perforans*, (**B1**,**B2**) *R. solanacearum* and (**C1**,**C2**) *P. syringae* pv. *tomato*.

**Figure 2 plants-12-03208-f002:**
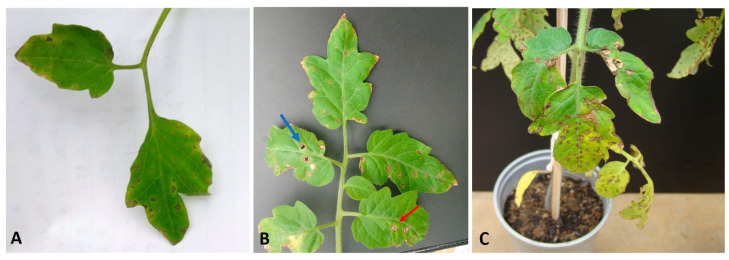
Efficacy of Gluconacin application in reducing the severity of bacterial spots caused by *X. perforans* on tomato plants. The disease symptoms in the leaves were assessed by visual estimation after (**A**) 7 DAI—Brown necrotic spots with yellowish halos on older leaves; (**B**) 10 DAI—Perforations caused by the detachment of the necrotic areas; (**C**) 16 DAI—Intensification of the chlorosis and coalition of perforations. In panel B, the red arrow marks characteristic bacterial spots, and the blue arrow indicates detachment from the necrotic area.

**Figure 3 plants-12-03208-f003:**
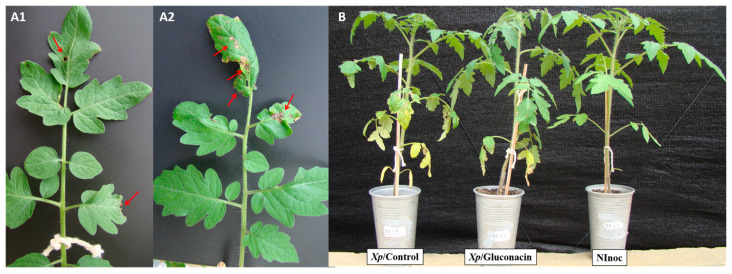
Reduced symptoms of bacterial spot disease caused by *X. perforans* in tomato plants treated with Gluconacin (**A1**) or untreated (**A2**). The arrows indicate perforations and necrotic areas. General aspect of non-inoculated plants (Ninoc) and inoculated with the pathogen that received (Xp/Gluconacin) or no (Xp/Control) bacteriocin treatment (**B**).

**Figure 4 plants-12-03208-f004:**
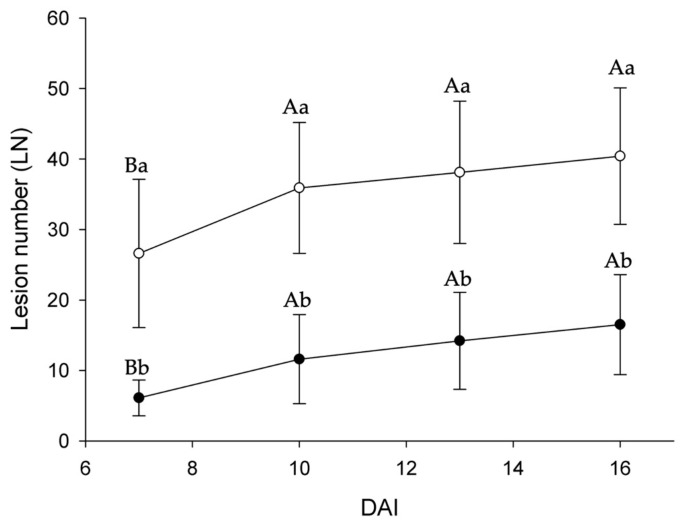
Gluconacin’s effectiveness reduced the number of lesions caused by bacterial spot disease over 16 DAI. The open dot (○) represents the control (lacking addition of Gluconacin) while the black dot (●) refers to the Gluconacin addition effect (1.0 μg·μL^−1^). Averages followed by identical capital letters do not differ in the DAI within each treatment. Averages followed by identical lowercase letters do not differ between treatments (control and Gluconacin). Bars represent the standard deviation.

**Figure 5 plants-12-03208-f005:**
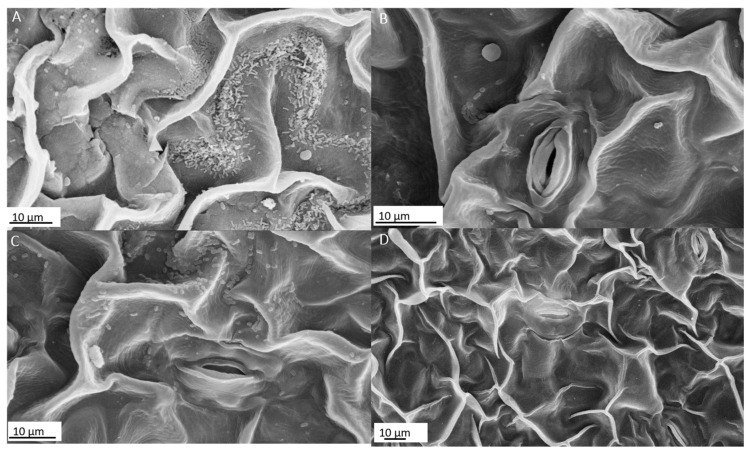
Scanning electron micrographs of tomato leaf samples from plants infected with *X. perforans* and untreated (**A**,**C**) or treated (**B**,**D**) with Gluconacin. The evaluations were made 24 h (**A**,**B**) and 72 h (**C**,**D**) after infection with the pathogen. The scale bars represent 10 μm.

**Figure 6 plants-12-03208-f006:**
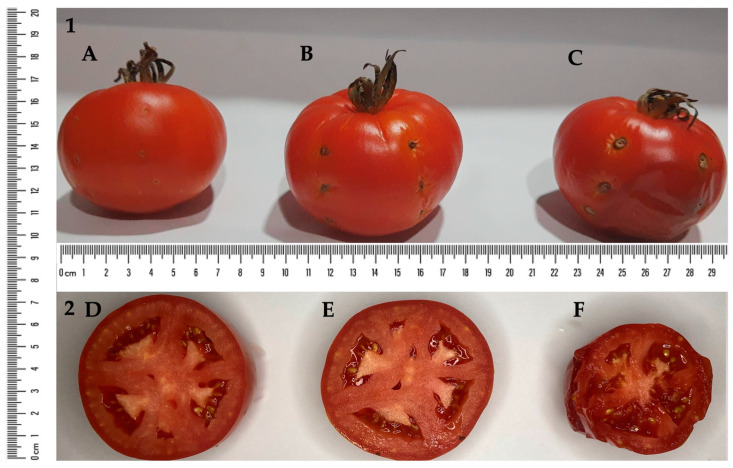
Effect of Gluconacin on tomato fruits. Disease symptoms in the fruits were assessed by visual estimation after 15 days. (1) Whole tomato fruits; (2) interior of tomato fruits, cut crosswise. (**A**,**D**), saline solution (0.85% NaCl); (**B**,**E**), elution buffer with *X. perforans* + Gluconacin; (**C**,**F**), elution buffer with *X. perforans*.

**Table 1 plants-12-03208-t001:** Severity (*AUDPC*) of bacterial spot disease in tomato plants treated with Gluconacin and the control’s elution buffer (Control I) and saline solution (Control II). Averages followed by identical lowercase letters do not differ between treatments (control and Gluconacin).

Treatments	NL	% Control
Gluconacin	111.3 ± 6.21 b	66.4
Control I	322.5 ± 4.64 a	-
Control II	331.1 ± 5.11 a	-
CV (%)	35.4	

Standard deviation (±) was calculated from the results of the three replicates.

## Data Availability

Not applicable.

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
