# Peer review of "The Effects of Gluconacin on Bacterial Tomato Pathogens and Protection against Xanthomonas perforans, the Causal Agent of Bacterial Spot Disease"

_plants, 2023, doi:10.3390/plants12183208_

Round 1

Reviewer 1 Report

The manuscript submitted by Elizabeth Teixeira de Almeida Ramos et al. reported that the bacteriocin Gluconacin inhibited the pathogenicity of R. solanacearum and Pseudomonas in vitro and showed efficient activity in reducing bacterial spots caused by X. perforans. Generally, the work displayed the broad spectrum bacteriocin activity in tomato plants. It is interesting and readable for the readers of the journal. However, the most important deficiency is lack of a regular applied commercial bacteriocin as a control in relative experiments including Fig 3 & 6, and Table 1.

 Others detail suggestions:

1. Title: effect to affects

   or: The effect of gluconacin on bacterial tomato pathogens and on protecting the plant against Xanthomonas perforans, the causal agent of bacterial spot disease

 1. Keep the correct format of bacterial species.

P. syringae pv. tomato, tomato should be italic also. P. syringae pv. tomato

X. oryzae pv. oryzae, oryzae should be italic also. X. oryzae pv. oryzae

Pseudomonas syringae pv. ciccaronei, italic ‘ciccaronei’

P. syringae subsp. savastanoi, italic ‘savastanoi’

X. vasicola pv. vasculorum italic ‘vasculorum’

2. give the statistical analysis in Fig 4

3. L233: “leading to impaired” to “impairing”

4. L268: ‘and bacterial leaf spot in’ to ‘of’

Author Response

Dear Reviewer #1,

The manuscript submitted by Elizabeth Teixeira de Almeida Ramos et al. reported that the bacteriocin Gluconacin inhibited the pathogenicity of R. solanacearum and Pseudomonas in vitro and showed efficient activity in reducing bacterial spots caused by X. perforans. Generally, the work displayed the broad spectrum bacteriocin activity in tomato plants. It is interesting and readable for the readers of the journal. However, the most important deficiency is lack of a regular applied commercial bacteriocin as a control in relative experiments including Fig 3 & 6, and Table 1 => We are grateful for your valuable suggestions and studies presented related with our work. Regarding the use of a commercial bacteriocin, the aim of our study was to prospect a novel bacteriocin from G. diazotrophicus, rather than comparing its applicability with other commercial bacteriocins. Furthermore, even if we wanted to perform a comparison, it wouldn't be feasible, as there is no bacteriocin available in the market that can achieve the desired spectrum of action, which would be: an antagonistic action against Xanthomonas perforans specifically in tomato.

  1. Title: effect to affects or: The effect of gluconacin on bacterial tomato pathogens and on protecting the plant against Xanthomonas perforans, the causal agent of bacterial spot disease. => the title has been improved, and it is now more meaningful: “The effect of gluconacin on bacterial tomato pathogens and on protecting the plant against Xanthomonas perforans, the causal agent of bacterial spot disease”.
  2. Keep the correct format of bacterial species. P. syringae pv. tomato, tomato should be italic also. P. syringae pv. tomato; X. oryzae pv. oryzae, oryzae should be italic also. X. oryzae pv. oryzae; Pseudomonas syringae pv. ciccaronei, italic ‘ciccaronei’; P. syringae subsp. savastanoi, italic ‘savastanoi’; X. vasicola pv. vasculorum italic ‘vasculorum’. => suggestion accepted.
  3. give the statistical analysis in Fig 4 => suggestion accepted. Statistical analysis performed and modified figure 4.
  4. L233: “leading to impaired” to “impairing”. => suggestion accepted.
  5. L268: ‘and bacterial leaf spot in’ to ‘of’. => suggestion accepted.

Reviewer 2 Report

The results presented here continue a recent series of work on bacteriocins. A selected one, Gluconacin, has been examined further, aiming for agricultural use. The work is well-planned, however, I see insufficiencies in every section of the MS. I discuss them in the returned PDF. The most severe issue is the SEM image showing the plant surface with and without the bacterial cells, depending on whether it was, or was not, treated with Gluconacin. The crinkled epidermal cells do not look as they should, and the complete absence of bacterial cells or cell debris needs further explanation. References do not match their citations, superficialness which rightly annoys reviewers. See further details in the returned PDF.

The MS is not far from satisfactory but would benefit from the work of an expert language editor. I have found strange-sounding words at several locations. Wrong word choices could distort the intended meaning.

Author Response

Dear Reviewer #2

The results presented here continue a recent series of work on bacteriocins. A selected one, Gluconacin, has been examined further, aiming for agricultural use. The work is well-planned, however, I see insufficiencies in every section of the MS. I discuss them in the returned PDF. The most severe issue is the SEM image showing the plant surface with and without the bacterial cells, depending on whether it was, or was not, treated with Gluconacin. The crinkled epidermal cells do not look as they should, and the complete absence of bacterial cells or cell debris needs further explanation. References do not match their citations, superficialness which rightly annoys reviewers. See further details in the returned PDF. => We are grateful for your valuable suggestions and studies presented related with our work. The sections considered incomplete in Materials and Methods have been considerably improved. Based on the circumstantial evidence from SEM micrographs presented in Fig A and C, we cannot state that the bacteria population attached to the leaf epidermal cell surface decreased from 24 to 72 hours. However, we should expect such an epiphytic population decline since it represents a transitory phase of the pathogen etiological infection process, where bacteria gain access to the inner tissue through the stomata aperture, colonize intercellular spaces and trigger changes in mesophyll cell that results in visual symptoms. I do not expect the same number of bacteria in treated leaves (Fig B compared to Fig A). Gluconacin application presumably kills or impairs the strong bacteria cell attachment process, which facilitates the removal of cell debris or bacteria cell weak associated with the SEM processing approach (wash steps and critical point drier). We could confirm it by using an environmental SEM. However, it is not so critical for the proposed work and objectives pursued. Regarding the flaws detected in the references, they were all corrected.

  • The MS is not far from satisfactory but would benefit from the work of an expert language editor. I have found strange-sounding words at several locations. Wrong word choices could distort the intended meaning. => An editorial office in Évora - Portugal (Georgia Bettine Morgan (UK passport 707852365)) edited the document, therefore its quality has improved and should be at a standard level of the Plants.
  • Manuscript - Content organization is not without redundant statements. The Methods part contains insufficient information for those who would want to repeat some experiments or for a reviewer to fully assess validity of data. Language: the MS would benefit from an expert editing work. => Given the above, we made significant changes throughout the manuscript. The sections considered incomplete in Materials and Methods have been considerably improved. An editorial office in Évora - Portugal (Georgia Bettine Morgan (UK passport 707852365)) edited the document, therefore its quality has improved and should be at a standard level of the Plants.
  • It would be informative to provide geographical data as the species are not worldwide individually => The suggestion was accepted and the requested information inserted in the text. “This disease has a generalized worldwide distribution and is caused by four Xanthomonas species: X. euvesicatoria and X. vesicatoria (distributed worldwide); X. gardneri (originally from Costa Rica and Yugoslavia and has been found in Brazil and USA) and X. perforans (USA, Mexico, Thailand, and Brazil) [8].”
  • where are the strains from? AND what peptone? => The suggestion was accepted and the requested information inserted in the text. “The bacterial strains used in the antagonism bioassays, Ralstonia solanacearum, Xanthomonas perforans and Pseudomonas syringae pv. tomato were obtained from the Phytopathogenic Bacteria Collection - Instituto Biológico de São Paulo and grown in Nutrient Broth medium (NB - 5.0 g.L–1 beef peptone; 3.0 g.L–1 beef extract; 5.0 g.L–1 NaCl, pH 7.0) at 30°C, 180 rpm for 24 hours.”
  • what species? AND correctly: lysogeny broth or simply LB - reference is missing.=> The suggestion was accepted and the requested information inserted in the text. “The production and purification of the bacteriocin Gluconacin was carried out as described by Oliveira et al. [19] with modifications. For heterologous expression, Escherichia coli strain BL21-AI™ was used, carrying the plasmid p17GDI0415N. Cultivation was performed in lysogeny broth medium (LB - 5.0 g.L–1 yeast extract, 10 g.L–1 NaCl; 10 g.L–1 tryptone, pH 7.0) [21] supplemented with 100 μg.mL–1 ampicillin, at 25°C, 180 rpm.”

  • A new method? (I assume yes.) How this can be related to the standard MIC method? AND what medium? what is the final concentration of cells in the assay? => The suggestion was accepted and the requested information inserted in the text. “The bacterial growth inhibited by the bacteriocin was submitted to a new assay evaluation to determine the MIC. The tests were performed as described by De Oliveira et al. [23], with a few modifications. The strains were grown to an optical density (OD600nm) of 0.5-0.8. Then, the cells were collected by centrifugation (4,000 rpm, 15 min) and resuspended in 100 µL of fresh LB medium (~108 CFU.mL–1), in a final concentration of (~104 CFU.mL–1. Subsequently, the purified Gluconacin was diluted in the elution buffer, and 100 µL of each dilution was added to the bacterial suspension, obtaining the following final concentrations of Gluconacin: 1.0; 0.5; 0.25; 0.125 and 0.06 µg.µL–1. The elution buffer was used as a control. The culture was maintained at 30°C, 180 rpm for 24 hours. Bacterial growth was monitored by counting the number of viable cells (CFU.mL–1).
  • where is this difference shown? (not in Fig which shows identical response by the strains) AND In MIC studies, a positive control (known antibiotic) is usually included. Here, where a new method for MIC determination was used, inclusion of a positive control is even more advised. => The higher sensitivity of Ralstonia solanacearum to bacteriocin Gluconacin is due to the fact that we used a lower concentration of this bactericion (0.5 µg.µL–1) and the effect was equal to that observed for X. perforans and P. syringae pv. tomato, where the concentration of 1.0 µg.µL–1 of gluconacin was used. In addition, a new table (containing MIC results) was inserted in the manuscript. The use of an antibiotic in the evaluation of MIC would first involve the selection of an antibiotic only to use it as a positive control in this evaluation. As this was not the focus of the work we did not find the inclusion of this step relevant.
  • please be more precise putting it. AND shown where? AND i.e., the plant? => The suggestion was accepted and the requested information inserted in the text. Changed figure 2B and legend of figure 2. In figure 2B we did not show the yellowing of the stem, but kind of symptom was observed. “The characteristic symptoms of bacterial spot disease were observed seven days after inoculation (DAI), and started in the older leaves, in the form of necrotic brown spots with yellowish halos (Figure 2A). Ten days after inoculation, some plants showed detachment from the necrotic area, resulting in perforations (Figure 2B), and exhibited yellowing of the stem (data not shown). Sixteen days after infection, intensification of chlorosis, coalescence of perforations and the fall of leaves were observed (Figure 2C). These symptoms were much more evident in plants inoculated and not treated with Gluconacin. The plants treated with Gluconacin showed a delay in the appearance of symptoms, in addition to being milder when compared to the untreated plants (Figures 3 and 4).” AND “In panel B, the red arrow mark characteristic bacterial spots and the blue arrow indicates the de-tachment from the necrotic area.”
  • There is a reduction in bacterial number from 24 to 72 hrs on untreated plants, which may mean that bacteria are unable to grow epiphytically. So, the many cells on the 24 hrs-plant are due their physical deposition by the spraying. The same amount of physical deposition is expected on Gluconacin-pretreated leaves. The lack of bacterial cells would mean that the deposited cells are removed from the surface without any trace (no corpses, remnants like ones shown by Ramos et al. 2022) by Gluconacin. How could that happen? You should provide explanation, arguments about the in vitro effect of the peptide (on bacterial cells only) and give data about what could happen to bacteria on control and peptide-pretreated leaves before 24 hrs (most importantly, do they multiply or not?) Another problem is that plant cells seem severely desiccated, (loss of cel turgor?). You'll need to replace these images with "proper" ones => Based on the circumstantial evidence from SEM micrographs presented in Fig A and C, we cannot state that the bacteria population attached to the leaf epidermal cell surface decreased from 24 to 72 hours. However, we should expect such an epiphytic population decline since it represents a transitory phase of the pathogen etiological infection process, where bacteria gain access to the inner tissue through the stomata aperture, colonize intercellular spaces and trigger changes in mesophyll cell that results in visual symptoms. I do not expect the same number of bacteria in treated leaves (Fig B compared to Fig A). Gluconacin application presumably kills or impairs the strong bacteria cell attachment process, which facilitates the removal of cell debris or bacteria cell weak associated with the SEM processing approach (wash steps and critical point drier). We could confirm it by using an environmental SEM. However, it is not so critical for the proposed work and objectives pursued. AND It can be a sample processing problem but consider that the bacteria colonization triggers stomata aperture leading to water loss in infected leaves.
  • In discussion, what do you mean? AND rewrite => The suggestion was accepted and the requested information inserted in the text. “The increasing population and global climate change are exercising enormous pressure on natural resources for food production. In addition, plant pests and diseases have a negative impact on agricultural production and consequently on the reduction of food for the human population [26]. In a world scenario in which 14% of the food produced is lost in stages prior to commercialization [27] it is crucial to search for more sustainable agricultural practices [28]. It is already known that more than 900 synthetic and natural antimicrobial peptides have been characterized and reported in the literature as potential strategies for agricultural use [29]. Bacteriocins have been highlighted in this category for their versatility of application in agriculture: performance on several phytopathogens [30], biostimulation of plant growth [31] and protection against abiotic stresses [32]. The use of agrochemical compounds conventionally used in disease and pest controls has been increasing considerably but consumer acceptance and desire for foods free from pesticide residues continues to grow [33].”
  • too cryptic to understand well. AND sounds like an unfinished sentence => Significant changes were made to the paragraph. “
  • these are strains, not molecules (bacteriocins)... AND as AU is not comparable, this statement is meaningless. => The content of the text was modified, we started to discuss with results of molecules. The article by Lavermicocca et al. it only presents data in Arbitrary Units (widely used in literature), so we decided to stick with the information presented.
  • how do we know it? AND With SEM, an almost complete disappearance of bacterial cells is seen, not just 66%. Cf also my previous comment on Fig. 5. AND some of these issues apply to bacteriocins too The biggest of them is the emerging resistance against Gluconacin, which must be explored in near future. You already showed that even strains of the same species respond very variably (Oliveira et al. 2018), so there are existing ways against this bacteriocin.=> The paragraph structure was modified, thus allowing a better understanding.

• In conclusion, how do you mean? => The phrase structure was modified. “In this manuscript, we demonstrated that the bacteriocin Gluconacin, derived from the PAL5 strain of G. diazotrophicus, showed promising biotechnological approach in the control of plant diseases, especially with regard to tomato pathogens.”

Round 2

Reviewer 1 Report

The manuscript had improved according to the comments. I think it can be accepted by the Journal.

Author Response

We are grateful for the careful review and believe that the article has been significantly improved following the reviewer's suggestions.

Reviewer 2 Report

- With Fig. 5 the problems remained. 1) The epidermal cells appear too much desiccated, presumably an artifact of sample preparation. 2) Not even dead bacterial cells are present on the gluconacin-treated plants. Both (1,2) phenomena are unexpected when we look at the other evidence you gave, so at least an attempt should be made to explain these - not in an answer letter to me, but in the manuscript. You say the SEM images are not critically important. However, I think all that you include must be there to strengthen your message. In summary, with the SEM more action is needed: either explain more or experiment more, e.g. try another technique like ESEM to see what happens really, or just to corroborate the existing SEM results.

- In 3.1, In my former suggestion I only wanted to see a better-phrased message on MIC values. I think the added (first) sentence and the table on MIC are not needed, is redundant. On the other hand, the lowest MIC (0.25 ug) for some strains is not mentioned and is missing. Please remove redundancy and complete the information about MICs, using only text (and Fig. 1).

- A positive control for the MIC study is to show readers that a known antibiotic is less or more effective against bacteria than gluconacin using the same evaluation method. It would provide a reference point and establish credibility in your results, a simple means to persuade readers that your compound is better than a usual one. And you say " this was not the focus of the work we did not find the inclusion of this step relevant".

"The article by Lavermicocca et al. it only presents data in Arbitrary Units (widely used in literature), so we decided to stick with the information presented." - Lavermicocca used a relativistic concentration in a context that provided reference points, points to be compared with. You did not provide such points, which is why "6,000 arbitrary units (AU)" says nothing.

I accept the other changes to the MS.

Author Response

Dear Reviewer #2,

We are grateful for your valuable suggestions and studies presented related with our work.

  1. With Fig. 5 the problems remained. 1) The epidermal cells appear too much desiccated, presumably an artifact of sample preparation. 2) Not even dead bacterial cells are present on the gluconacin-treated plants. Both (1,2) phenomena are unexpected when we look at the other evidence you gave, so at least an attempt should be made to explain these - not in an answer letter to me, but in the manuscript. You say the SEM images are not critically important. However, I think all that you include must be there to strengthen your message. In summary, with the SEM more action is needed: either explain more or experiment more, e.g. try another technique like ESEM to see what happens really, or just to corroborate the existing SEM results. ==> Dear reviewer, we made a mistake when indicating the inoculated leaves treatment in the text (line 235 – the correct is Figure A and C) for 24 and 72 hours after X. perforans foliar spraying. Please consider the newest version of this section (lines 233 to 240). Also, please see lines 323 to 328. As quoted, the epidermal cells' appearance can combine cell preparation problems under critical point drier processes and part of the bacterial etiological cycle that modulates stomata aperture, increasing tissue water loosening. Even considering this, the epidermal cell attachment process was not compromised, evidencing that the foliar spray was adequate for the pathogen's phylloplane establishment. Information was also included in the results and discussion, additionally explaining the findings and the quality of the images.
  2. In 3.1, In my former suggestion I only wanted to see a better-phrased message on MIC values. I think the added (first) sentence and the table on MIC are not needed, is redundant. On the other hand, the lowest MIC (0.25 ug) for some strains is not mentioned and is missing. Please remove redundancy and complete the information about MICs, using only text (and Fig. 1) ==> We agree that the first sentence above Table 1 and Table 1 itself were repetitive. We followed the suggestion to remove them from the text. Consequently, we chose to retain Table 1 as supplementary material (Table S1), as it displays the standard deviations of the mean MICs for each pathosystem along with the mean comparison test, highlighting the statistical difference between concentrations. Additionally, we addressed the request by incorporating the minimum inhibitory concentration (MIC) of gluconacin into the text.
  3. A positive control for the MIC study is to show readers that a known antibiotic is less or more effective against bacteria than gluconacin using the same evaluation method. It would provide a reference point and establish credibility in your results, a simple means to persuade readers that your compound is better than a usual one. And you say " this was not the focus of the work we did not find the inclusion of this step relevant". ==> When conducting laboratory MIC assays to evaluate the efficacy of bacteriocins, it is important to justify the decision not to use antibiotics as positive controls. This choice is based on several critical factors: 1- Differences in Mechanisms of Action: Bacteriocins and antibiotics could have distinct mechanisms of action. Using antibiotics as positive controls can yield misleading results as the modes of action of bacteriocins may significantly differ. This could lead to misinterpretations of the actual effectiveness of bacteriocins; 2- Resistance Development: Including antibiotics as positive controls might promote the development of antibiotic-resistant bacteria, not accurately reflecting bacteriocins' ability to address bacterial resistance. Bacteriocins might have different implications for resistance development, making the use of antibiotics inadequate for understanding this aspect; 3- Effects on Microbial Ecosystem: Agricultural environments are complex ecosystems with interactions among various microorganisms. The presence of antibiotics as positive controls could disrupt this delicate ecology, affecting both pathogens and beneficial microorganisms. This could lead to undesired ecological impacts and affect overall soil and plant health; 4- Specific Sensitivity of Bacteriocins: Bacteriocins often have more limited spectra of activity compared to antibiotics. The use of antibiotics as positive controls might not accurately predict the specific sensitivity of bacteria to bacteriocins' effects, which can differ in their ability to target certain species; 5- Realistic Assessment of Bacteriocins: Using antibiotics as positive controls can lead to misconceptions about the effectiveness of bacteriocins in real field situations. Bacteriocins might possess unique properties that make them more effective or safer for agricultural use compared to conventional antibiotics; 6- Public Health and Environmental Concerns: Excessive antibiotic use in agriculture has raised concerns about antibiotic resistance and residues in food. Utilizing bacteriocins instead of antibiotics reflects an approach more aligned with sustainable and responsible practices expected by society and regulations. In summary, when conducting MIC assays with bacteriocins in the laboratory, the decision not to employ antibiotics as positive controls is essential to ensure more accurate and relevant results for the application of bacteriocins in agricultural contexts. Our MIC assay was based on previous studies conducted in a similar manner, published by: Soltani S, Biron E, Ben Said L, Subirade M, Fliss I. Bacteriocin-Based Synergetic Consortia: a Promising Strategy to Enhance Antimicrobial Activity and Broaden the Spectrum of Inhibition. Microbiol Spectr. 2022 Feb 23;10(1):e0040621. ; V A L, Mohammed Alarjani K, Malaisamy A, Balasubramanian B. Bacteriocin producing microbes with bactericidal activity against multidrug resistant pathogens. J Infect Public Health. 2021 Dec;14(12):1802-1809.
  4. "The article by Lavermicocca et al. it only presents data in Arbitrary Units (widely used in literature), so we decided to stick with the information presented." - Lavermicocca used a relativistic concentration in a context that provided reference points, points to be compared with. You did not provide such points, which is why "6,000 arbitrary units (AU)" says nothing.==> Following the reviewer's suggestion, we have replaced the reference with a more coherent one.
  5. I accept the other changes to the MS. ==> We are grateful for the careful review and believe that the article has been significantly improved following the reviewer's suggestions.
